

# A measurement system for vertical seawater profiles close to the
# air/sea interface
**Richard P. Sims[1,2], Ute. Schuster[2], Andrew. J. Watson[2], Ming Xi. Yang[1], Frances. E.**
**Hopkins[1], John. Stephens[1] and Thomas. G. Bell[1,*]**
[1]{Plymouth Marine Laboratory, Plymouth, United Kingdom}
[2]{University of Exeter, Exeter, United Kingdom}
*Correspondence to:* T. Bell (tbe@pml.ac.uk)
**Abstract**
This paper describes a Near Surface Ocean Profiler, which has been designed to precisely
measure vertical gradients in the top 10 m of the ocean. Variations in the depth of seawater
collection are minimised when using the profiler compared to conventional CTD/rosette
deployments. The profiler consists of a remotely operated winch mounted on a tethered yet
free floating buoy, which is used to raise and lower a small frame housing sensors and inlet
tubing. Seawater at the inlet depth is pumped back to the ship for analysis. The profiler can be
used to make continuous vertical profiles or to target a series of discrete depths. The profiler
has been successfully deployed during wind speeds up to 10 m s$^{-1}$ and significant wave
heights up to 2 m. We demonstrate the potential of the profiler by presenting measured
vertical profiles of the trace gases carbon dioxide and dimethylsulfide. Trace gas
measurements use an efficient microporous membrane equilibrator to minimise the system
response time. The example profiles show vertical gradients in the upper 5 m for temperature,
carbon dioxide and dimethylsulfide of 0.15 °C, 4 μatm and 0.4 nM respectively.

**1    Introduction**
Exchange between the ocean and atmosphere is an important process for many gases.
Important examples include carbon dioxide ($CO_2$), for which the oceans account for 25% of





the sink for anthropogenic emissions (Le Quéré et al., 2016), and dimethylsulfide (DMS),
which has an oceanic source and influences cloud properties with implications for the global
energy balance  (Quinn and Bates, 2011). The magnitude and direction of air/sea gas transfer
is typically represented by Flux=$K\Delta C$ (Liss and Slater, 1974), where $\Delta C$ is the concentration
difference across the air-sea interface and $K$ is the gas transfer velocity. Direct flux
measurements (Bell et al., 2013; Yang et al., 2013; Miller et al., 2010) are only possible for a
small number of gases and are not made routinely.  Most flux estimates use a wind speed-
based parameterisation of $K$ (e.g. Wanninkhof, 2014) coupled with measurements of $\Delta C$.
$CO_2$ is the most well-observed trace gas in the surface ocean, with 14.5 million measurements
compiled into a global database, the Surface Ocean $CO_2$ Atlas (SOCAT),
http://www.socat.info/ (Bakker et al., 2016). Global trace gas databases also exist for gases
such as methane and nitrous oxide https://memento.geomar.de/ (Bange et al., 2009),
dimethylsulfide http://saga.pmel.noaa.gov/dms/ (Lana et al., 2011) and halocarbons
https://halocat.geomar.de/ (Ziska et al., 2013). Accurate estimation of air/sea flux requires
concentration measurements that are representative of the interfacial concentration difference.
Surface seawater samples are often collected from the underway seawater intake of research
vessels, typically at 5-7 m depth. A source of potential error in air/sea flux calculations arises
from the assumption of vertical homogeneity within the mixed layer (Robertson and Watson,
1992). If vertical concentration gradients exist in the mixed layer, then underway seawater is
not representative of the interfacial layer, which could create a global sampling bias (McNeil
and Merlivat, 1996).
Vertical gradients in trace gas concentrations have been observed under conditions that are
favourable for near surface stratification (Royer et al., 2016). At low wind speeds, high solar
irradiance can suppress the depth of shear-induced mixing to create a near surface layer
several degrees warmer than the water below (Ward et al., 2004; Fairall et al., 1996). Near
surface stratification in the marine environment can also be induced by freshwater inputs such
as rain (Turk et al., 2010) and riverine discharge. Changes in surface seawater temperature
and salinity alter the solubility of dissolved gases and thus the amount available for air/sea
exchange (Woolf et al., 2016). Dissolved gases isolated in the upper few metres of the ocean
may additionally be modified by physical process such as air/sea exchange and
photochemistry. Marine biota confined within the stratified layer (Durham et al., 2009), may


also alter trace gas concentrations. For the purposes of this paper, near surface gradients are
defined as physical and/or chemical gradients in the upper 10 m of the ocean.
Identifying and quantifying near surface gradients in trace gas concentrations is challenging.
Ship motion often inhibits near surface measurements made with the standard oceanographic
approach of sampling with Niskin bottles mounted on a CTD rosette. Substantial vertical
movement of the rosette limits how close to the surface a sample can be taken. For example, a
crane arm 4 m above the sea surface and 11 m from the centreline of a ship that is rolling by
±4 degrees will induce ~1.5 m sample depth variation every few seconds. CTD/Niskin bottle
sampling requires that the rosette is kept below the sea surface. Sampling within 2 m of the
sea surface is often impossible, even under relatively calm conditions.
We present a Near Surface Ocean Profiling buoy (NSOP) designed for measuring near surface
profiles. The design principles for NSOP were:
(1) Platform diameter less than the wavelength of most open ocean waves, allowing it to ride
the swell;
(2) Short sampling arm close to the sea surface to reduce vertical movements induced by
platform motion;
(3) Capable of deployment close to the ship (to retrieve water for trace gas analysis), but away
from major turbulence and motion due to the ship itself.
Example profiles from a cruise on the European continental shelf (*RRS Discovery*, DY033,
July 2015) and in the English Channel on board the *RV Plymouth Quest* (part of the Western
Channel Observatory, Smyth et al., 2010, April 2014) are discussed.
**2    Methods**
**2.1    Near Surface Ocean Profiler (NSOP) description**
NSOP is a repurposed ocean buoy (1.6 m diameter) with a central lifting eyelet (Fig. 1). The
top of the buoy is 0.5 m above the sea surface. Mounted on top of the buoy are a line of sight,
remotely operated winch (Warrior Winch, model C8000) and a gel battery (Haze, model
HZY-S112-230). The winch feeds Kevlar rope through a block and tackle with a 3:1 ratio to
reduce rope pay-out speed to ~0.05 m s$^{-1}$. The block and tackle is attached to the end of an





outstretched arm 0.25 m from the outer edge of the buoy. The winch line is attached to an
open frame (0.35 m diameter, 0.8 m height) with the capacity to house multiple sensors.
Desired sampling depth is targeted using knowledge of the winch pay-out speed. Rope pay-
out is then timed with a stopwatch. This approach only approximately regulates the sampling
depth because: (i) winch pay-out varies slightly depending on the amount of rope on the
spool; and (ii) variable horizontal current strength affects the vertical versus horizontal
position of the sampling frame. To minimise horizontal movement of the sampling frame we
attached a 10 kg weight to the base of the frame.
The primary sensor on the sampling frame is a small CTD (Valeport miniCTD) set to sample
at a high frequency (>1 Hz). Under calm conditions it is possible to sample as close as 0.1 m
from the air/sea interface when the miniCTD and tubing are mounted near the top of the
frame. Rougher conditions demand that the frame be kept deeper (~0.5 m) as motion can
momentarily bring the sensors and tubing out of the water. An emergency tag line was
attached to the sampling frame in case the winch line failed. Seawater for trace gas analysis
was pumped back to the ship at 3.5 L min$^{-1}$ through a 50 m PVC hose (0.5 in ID). A heavy
duty peristaltic pump (Watson Marlow, model 701IB/R), primed with water from the ships
underway supply was used to overcome the large hydraulic head (~4 m). The open end of the
tubing was located at the same depth as the miniCTD. Water arriving to the ship's laboratory
was divided, with ~3.0 L min$^{-1}$ for flow-through analysis (e.g. equilibrator for trace gases) and
~0.5 L min$^{-1}$ for discrete samples (e.g. total alkalinity).
We assessed the depth resolution capability of NSOP at a particular depth by looking at
pressure variations under calm conditions with a fixed amount of winch rope paid out. In calm
to moderate conditions (<2.5 m significant wave height) the amount of vertical movement
indicated by the standard deviation (SD) in the depth is ±0.18 m (see Fig. S2 in Supplemental
information). During 4 deployments in rough conditions (>2.5 m significant wave height), the
depth variability increased as the sampling frame was lowered (at 5 m, SD was ±0.275 m).
**2.2   NSOP deployment**
On a large research vessel such as the *RRS Discovery*, the deployment and recovery of NSOP
requires close coordination between the bridge and 3 personnel on deck. NSOP was always
deployed while the ship was on station. Ship orientation during deployments was typically
with bow into the wind but also accounted for swell and current direction/speed. NSOP was



lifted by the aft crane (Fig. 1). Once NSOP was lowered to the surface it was detached from
the crane via a quick release. Two slack lines were looped through eyelets on the free-floating
NSOP to maintain its position close to the ship. A third slack line was connected to the top of
the buoy and passed through a block on a crane arm to maintain distance (minimum 7 m)
between NSOP and the ship. The slack lines successfully inhibited the tendency of NSOP to
drift horizontally without disrupting its ability to ride the swell. The instrument frame acted
like a sea anchor and minimised rotation of NSOP. A 4 m lifting strop used for recovery was
connected to the lifting eyelet and loosely lashed to the aft slack line. During retrieval, the
slack lines were hauled in and the crane and jib arms brought towards the ship to bring NSOP
alongside. The lifting strop was then parted from the slack line and attached to the crane to lift
NSOP back on deck.  For photographs of each stage of a NSOP deployment and videos of a
deployment and in operation, see supplemental material.
Turbulence from the ship's propellers has the potential to mix the water column and destroy
any near surface gradients. The ship did not use the aft thrusters whenever conditions were
suitable (mild sea state, weak currents and no local hazards). Keeping NSOP away from the
ship limited disruption of near surface gradients by the thrusters and reduced the risk of line
entanglement in the aft propellers.  Our winch did not have a groove bar to feed the rope onto
the winch drum, leading to an increased likelihood of snagging during spooling. To minimize
snagging, the rope was manually fed onto the winch spool before deployments.  Visual
monitoring of the NSOP frame, slack lines and winch spool is important during deployment.
NSOP has been successfully deployed in 'moderate' sea states up to Beaufort force 5 (~10 m
s$^{-1}$ wind speed and wave heights of ~2.0 m). Deployment length typically varied from 1-3
hours.



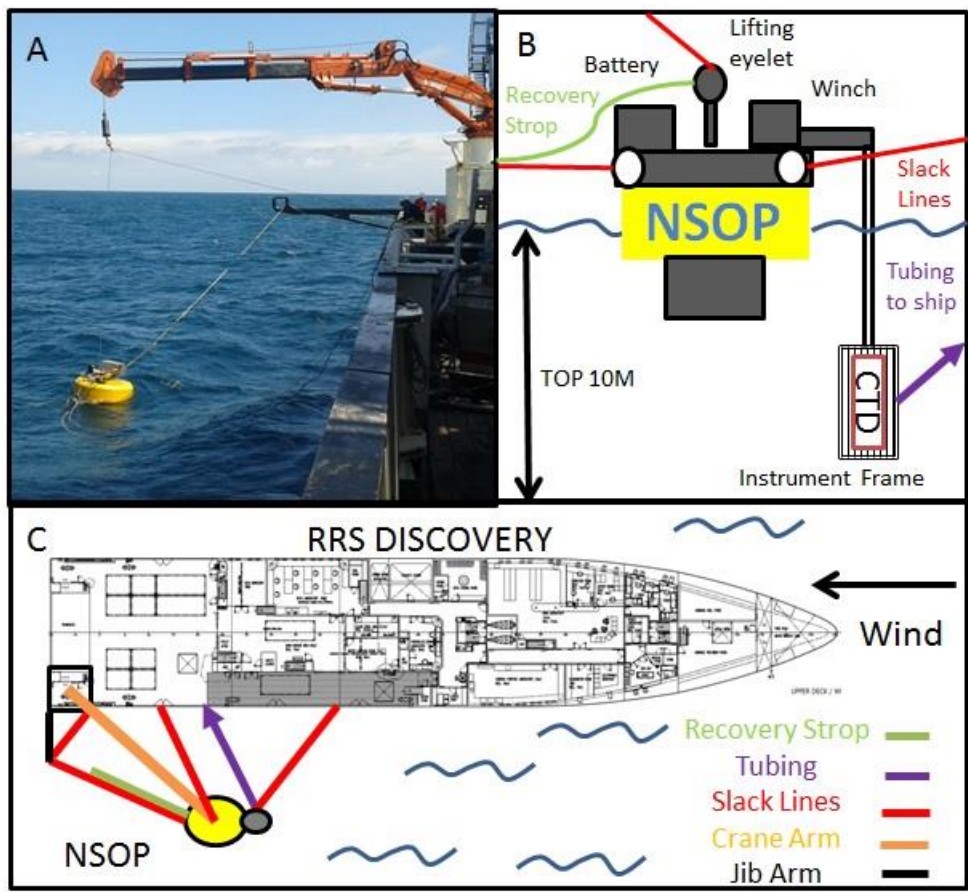

**Figure 1:** Different points of view of an NSOP deployment: (a) Image from a deployment on *RRS Discovery* in May 2015 (Cruise DY030); (b) Schematic cross section of NSOP including tubing back to ship (purple) and slack lines (red); and (c) Top down schematic from a research ship including ship orientation. Not to scale.

NSOP can be used in two profiling modes: 'continuous' and 'discrete'. Continuous profiling maximises vertical coverage and involves the winch continuously paying rope in and out at ~0.05 m s$^{-1}$. A complete down/up profile to 10 m can be conducted in approximately 7 min (Fig. 9). Depth resolution during continuous profiling is determined by the measurement response time. Instruments with rapid response times such as the miniCTD temperature and conductivity sensors (0.15 s and 0.09 s) have theoretical depth resolutions of 0.75 cm and 0.45 cm respectively. Actual depth resolution will also be affected by the sampling depth



variability of the NSOP instrument frame. A measurement setup with a longer response time
(such as for seawater $CO_2$) requires a different approach (see Section 2.5).
During discrete profiling, the winch pays out a fixed amount of rope (typically 0.5 m) and the
sampling frame is left at a fixed depth. After a fixed sampling period, more rope is paid out.
The process is repeated down and then up such that a set of discrete depths are sampled in a
'stepped' profile. The discrete profiling depth resolution is determined by the depth
fluctuations when sampling at a fixed depth (see Section 2.1). Discrete profiles are a more
appropriate approach for measurement systems with a longer response time. A discrete profile
with 0.5 m steps down to 5 m and back to the surface using a 2.5 min sampling period takes
about an hour. The sampling period at each depth and frequency/distribution of depths within
the profile can be adjusted to suit sampling priorities.
The maximum deployment time is limited by the capacity of the winch battery. When under
no load, the battery allows for approximately 3 hours of operation in the continuous mode.
Discrete profiling requires substantially less winch usage such that battery drainage is even
less of a concern.
## 2.3   $CO_2$ analysis
The $CO_2$ measurement system (Fig. 2) is a modified version of the system described by
(Hales et al., 2004). Seawater from the NSOP inlet was passed through the equilibrator (see
Section 2.3.1) at ~3 L min$^{-1}$ and the flow rate monitored (Cynergy ultrasonic flow meter,
model UF25B). A compressed nitrogen gas supply, maintained at a constant flow rate of 100
ml min$^{-1}$ (Bronkhurst mass flow controller, model F-201-CV-100) flows through the
equilibrator in the opposite direction to the seawater flow. The gas has high water vapour
content after equilibration and is dried (Permapure nafion dryer, model MD-110-48S-4). The
dried sample then enters the analytical cell of a NDIR Licor 7000, which is protected with a
0.2 µm filter (Pall, Acro 50).
$CO_2$ measurements at atmospheric pressure as recommended by Dickson et al. (2007) were
not possible due to the nature of the experimental setup. The continuous gas flow through the
system caused a small 0.4 kPa pressure increase in the Licor measurement cell. The elevated
pressure was taken to be representative of the equilibrator pressure and was used to obtain the
partial pressure of $CO_2$ in the equilibrator ($pCO_{2(eq)}$).



The Licor was calibrated using three $CO_2$ standard gases before and after each NSOP
deployment. The concentrations of the standard gases (BOC Ltd.) were determined by
referencing against US National Oceanic and Atmospheric Administration certified standards
(244.91, 388.62, 444.40 ppm) in the laboratory. The seawater temperature at the entry and
exit ports of the equilibrator was recorded at 1 Hz (Omega ultra-precise 1/10 DIN immersion
RTD) using stackable microcontrollers (Tinkerforge master brick 2.1 and PTC bricklet).
Equilibrator temperature probes and the miniCTD temperature sensor were calibrated before
and after each cruise against an accurate reference sensor (Fluke, model 5616-12, ±0.011°C)
in a stable water bath (Fluke 7321).

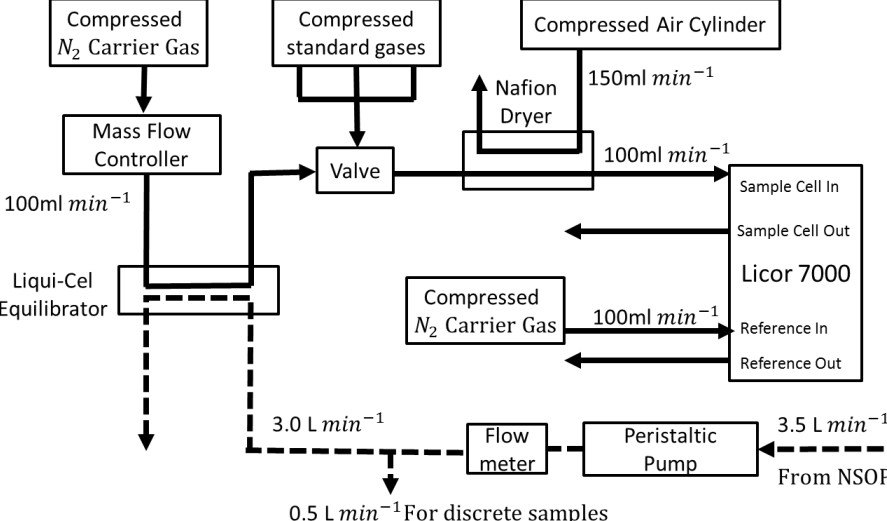

**Figure 2:** $CO_2$ system schematic. Solid and dashed arrows correspond to gas and water flows
respectively. The Licor reference cell is flushed with equilibrated gas at 100 ml min[-1]. A
manual selection valve was used to switch between equilibrated gas and the $CO_2$ standards.
2.3.1   Equilibrator
The showerhead equilibrator is the most commonly-used equilibrator for $CO_2$ but takes ~100
s to equilibrate (Dickson et al., 2007; Kitidis et al., 2012; Körtzinger et al., 2000; Webb et al.,
2016). This equilibration time is too slow for effective use during NSOP deployments. We



used a membrane equilibrator (Liqui-Cel, model 2.5x8) because it has a large surface area to
volume ratio and membrane porosity (50%). The Liqui-Cel expedites gas transfer and
efficiently achieves equilibration (Loose et al., 2009), with a 3 s response time for $CO_2$ (Hales
et al., 2004). Membrane equilibrators have been used by others for trace gas analysis (Hales et
al., 2004; Marandino et al., 2009).
Fugacity of seawater $CO_2$ is calculated from the Licor gas phase $CO_2$ measurement. This
approach assumes that the gas phase sample has equilibrated fully with the seawater. We
performed equilibration efficiency experiments in a seawater tank using a showerhead
equilibrator as a reference. Liqui-Cel equilibration efficency declined after prolonged
exposure to seawater, likely due to biofouling of the membranes. In a fouled equilibrator,
equilibration efficency was a function of the flow rate on both the water and gas side of the
membrane. An increased gas flow rate reduces the residence time inside the Liqui-Cel and
allows less time to equilibrate (Fig. 3a). Increasing the waterside flow rate moves the gas
phase closer to equilibrium because the transfer coefficent in the membrane increases (Fig.
3b).
Cleaning with an acid - base sequence restored the efficiency of a fouled equilibrator. It was
necessary to actively pump chemicals through the Liqui-Cel to achieve a full recovery in
efficiency. For more details on cleaning techniques, see supplemental material. Efficiency
reductions in membrane equilibrators like the Liqui-Cel have not been reported by previous
studies. Some authors have used 5-50 μm filters to minimise biofouling (Hales et al., 2004)
but this was not possible with the NSOP experimental design. If filtering seawater is not
possible, we recommend flushing with freshwater after use, regular cleaning of the Liqui-Cel
and daily tests to quantify equilibration efficiency. Trace gas measurement systems that use
an internal liquid phase standard (e.g. dimethylsulfide, Section 2.4) account for any changes
in equilibrator efficiency.





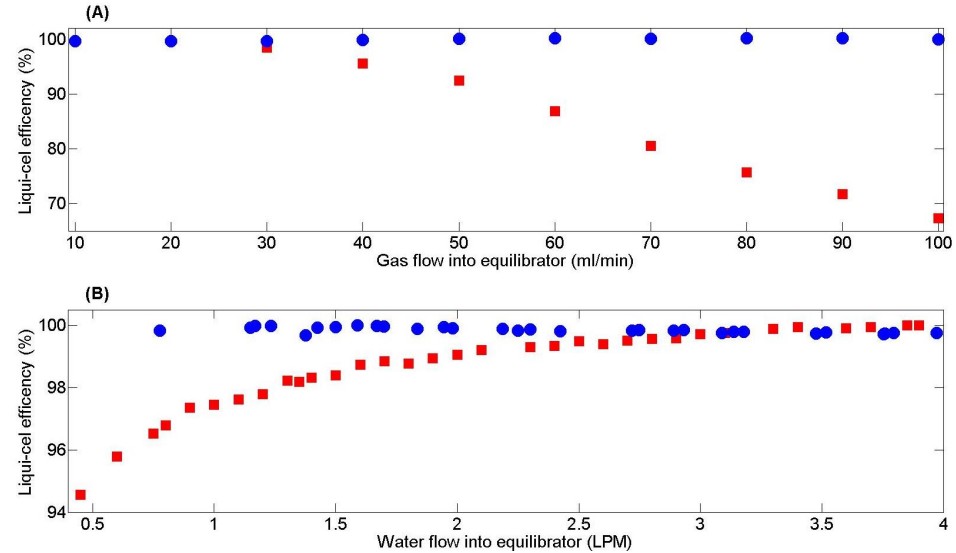

**Figure 3:** Liqui-Cel $CO_2$ equilibration efficiency (Liqui-Cel mixing ratio / showerhead
mixing ratio) for: (a) changing gas flow at a fixed water flow rate of 4 L min$^{-1}$; and (b)
changing water flow  at a fixed gas flow of 100 ml min$^{-1}$. Blue = unfouled equilibrator. Red =
fouled equilibrator.
**2.4   DMS analysis**
DMS was measured with Atmospheric Pressure-Chemical Ionisation Mass Spectrometry
(API-CIMS), using a system modified following Saltzman et al. (2009). Measurements were
calibrated using an isotopic liquid standard of tri-deuterated DMS (see Bell et al., 2013 for
details).  Isotopic standard was injected at 120 μL min$^{-1}$ into the 3 L min$^{-1}$ seawater flow from
NSOP before it entered the Liqui-Cel equilibrator. Compressed nitrogen gas was passed
through the equilibrator in the counter direction to the seawater flow at 1 L min$^{-1}$.  The use of
an internal standard meant that any incomplete equilibration of the ambient non-isotopic DMS
was also true for the isotope. The gas stream exited the equilibrator and was dried (Permapure
nafion dryer, model MD-110-48S-4) before entering the mass spectrometer for analysis. DMS
was detected at m/z 63 and the isotopic standard detected at m/z 66. The concentration of
DMS was calculated using the ion signals and relevant flow rates (Bell et al., 2015). This
approach has been shown to compare well with other analytical techniques for DMS (Walker
et al.., 2016; Royer et al., 2014).



**2.5  NSOP response and delay time**
The response time of the NSOP setup was determined by simulating step changes in gas
concentrations. The tubing inlet was quickly transferred between two buckets of seawater
with a distinct difference in concentration. A model fit to the exponential change in signal was
used to estimate the response time (Fig. 4). We estimate the system response time (e-folding
time) for $CO_2$ as 24 s, which is slightly faster than the 34 s reported by Webb et al. (2016).
The e-folding time in the DMS signal is estimated as 11 s, which is consistent with the rapid
gas flow rate through the analytical system.
Continuous profiling with the $CO_2$ system and a 24 s response time yields a depth resolution
of 1.2 m, which is greater than the required resolution to assess near surface gradients. DMS
has a faster response time than $CO_2$, but in continuous profiling mode this only translates to a
depth resolution of 0.6 m, slightly less than the 1.2-2 m reported by (Royer et al., 2014). A
depth resolution of $< 0.5$ m was desired to capture upper ocean vertical gradients in $CO_2$ and
DMS so NSOP was operated in discrete profiling mode.
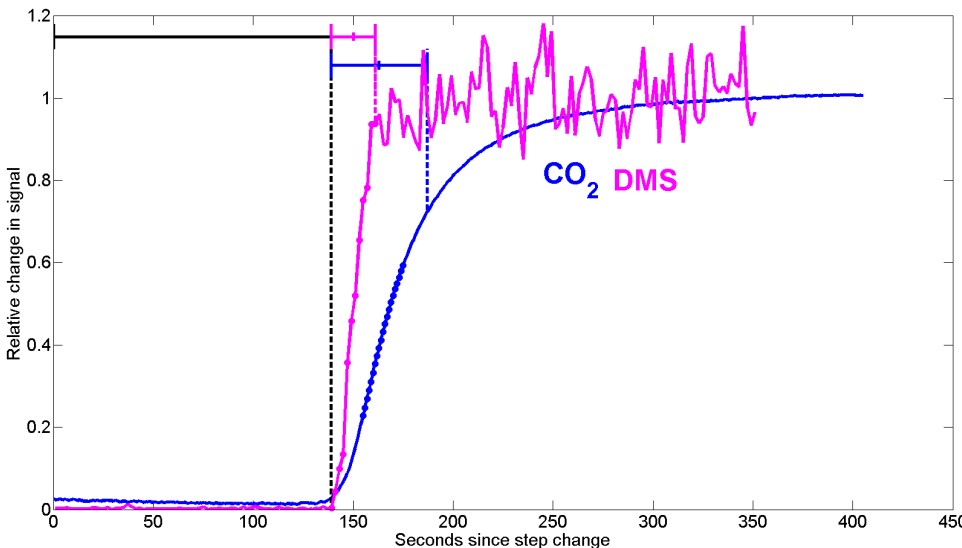
**Figure 4:** Instrument responses to step changes in seawater $CO_2$ (blue) and DMS (magenta).
Instrument responses have been scaled so that the initial and end concentrations are between 0
and 1. Time is referenced against the point when the step change was initiated. The response
is seen in both instruments after a delay of 139 s (black dashed line). Two e-foldings are
indicated by vertical dashed lines for $CO_2$ (blue) and DMS (magenta). The data points marked
by circles were used to make an exponential fit to the data to determine the response time
(Sect 2.5).
We used different approaches to assess the delay between instantaneous miniCTD
measurements and water arriving to the ship for analysis. Using the internal volume of NSOP
tubing (0.5 in ID, 54 m length) and a seawater flow rate of 4.15 L min$^{-1}$, the tubing delay to
the equilibrator was calculated as 114 s. Delay correlation analysis between the NSOP
miniCTD temperature and a temperature sensor positioned at the entrance to the equilibrator
suggests a delay of 112 s. The delay between a bucket switch and a $CO_2$ change in the Licor
was timed at 138 s. The bucket switch delay was longer because the bucket switch experiment
also accounts for the delay in the equilibrator and the Licor.
**2.6   Data processing**
During discrete profiling, distinct sample depths were identified from the rapid changes in
pressure during depth transitions. Data were binned into discrete depth bins using CTD
pressure measurements. Trace gas data were assigned to depth bins after adjusting for the
calculated transit time through the NSOP tubing (Section 2.5). $CO_2$ data from the beginning
(2 e-foldings + 15 s buffer = 63 s) and end (15 s buffer) of each depth bin was excluded from
analysis to account for the response time of the system and the transition time between sample
depths. The same approach was taken for DMS, where the faster response time resulted in a
smaller portion of data excluded at the beginning of each depth bin (2 e-foldings + 15 s buffer
= 37 s).
The $CO_2$ mixing ratio ($xCO_2$) measured in the Licor is converted to equilibrator fugacity
($fCO_{2(eq)}$) using calibration standards, *in situ* seawater salinity, and the pressure and
temperature in the equilibrator (SOP 5# Underway pCO2Dickson et al., 2007). Vertical
profiles of seawater $CO_2$ fugacity ($fCO_{2(sw)}$) are calculated using average equilibrator fugacity
($fCO_{2(eq)}$), equilibrator temperature ($T_{(eq)}$) and *in situ* seawater temperature ($T_{(sw)}$) at each
depth (Takahashi et al., 1993). The time series $fCO_{2(sw)}$ data shown in (Fig. 7) are also
calculated using the same equation from Takahashi et al. (1993) but instead use high
frequency $fCO_{2(sw)}$, $T_{(eq)}$ and $T_{(sw)}$ data.



## 2.7  Seawater sample collection using NSOP
The NSOP setup enables vertical profiles of discrete seawater samples to be collected from
upstream of the equilibrator, with a split in the tubing diverting ~0.5 L min$^{-1}$ into a sink. For
example, discrete seawater samples (250 ml) have been successfully collected and analysed
for Total Alkalinity (TA). Samples were collected and poisoned following best practice
recommendations (SOP#1, (Dickson et al., 2007). Bottle filling plus 1 overfill took ~60 s.
Start and end times were recorded so that collection depth could be retrospectively
determined from the CTD pressure data. Analytical methods and an example depth profile
(Fig. S3) are provided in Supplementary information.
## 3  Field Measurements / Observations
Presented below are example profiles collected using NSOP. The first deployment was in the
open ocean (July 30[th] 2015, Central Celtic Sea; 49.4213°N, -8.5783°E) from the *RRS*
*Discovery* (100 m length, 6.5 m draught). The second deployment was in coastal waters (15[th]
April 2014, Plymouth Sound; 50.348°N, -4.126°E) from the *RV Plymouth Quest* (20 m
length, 3 m draught).
### 3.1  Open ocean deployment
NSOP was deployed at 14:05 (UTC) on 30[th] July 2015. During the 6 hours preceding
deployment, the ship was on station and encountered persistently strong solar radiance (> 600
W m$^{-2}$), mild winds (< 6 m s$^{-1}$) and calm sea state (significant wave height < 1.6 m). This
combination of low wind speeds and high irradiance (Fig. 5) is favourable for near surface
stratification (Donlon et al., 2002).



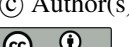

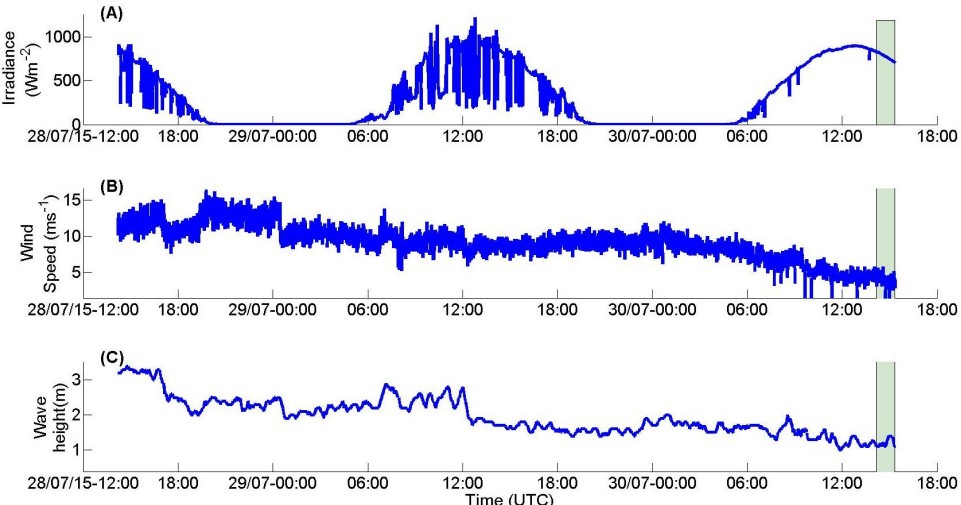

**Figure 5:** Timeseries of meteorology and sea state variables in the Celtic Cea in July 2015
while the ship was on station: (a) irradiance; (b) wind speed; and (c) significant wave height.
The data begin 48 h before the start of the profile at 14:05 hrs (UTC). The vertical grey bar
indicates the period when NSOP was profiling.
Fig. 6 presents the time series data collected by NSOP for depth, temperature, salinity and
$fCO_{2(sw)}$. Discrete profiling began at 14:05 hrs (UTC) at 0.7 m depth, which was as close to
the surface as the frame could be located without the possibility of breaking the surface.
Depth bins were identified based on rapid depth transitions (Fig. 6a). Bottles were filled for
discrete samples during the downcast. Profiling lasted 75 minutes and finished back at the
surface at 15:20 hrs (UTC). Seawater temperature was 16.61± 0.06 °C and $fCO_{2(sw)}$ was
undersaturated with respect to the atmosphere; both were the expected magnitude for summer
in the Celtic Sea (Frankignoulle and Borges, 2001). Salinity was homogeneous throughout the
NSOP deployment, only varying by ±0.004.



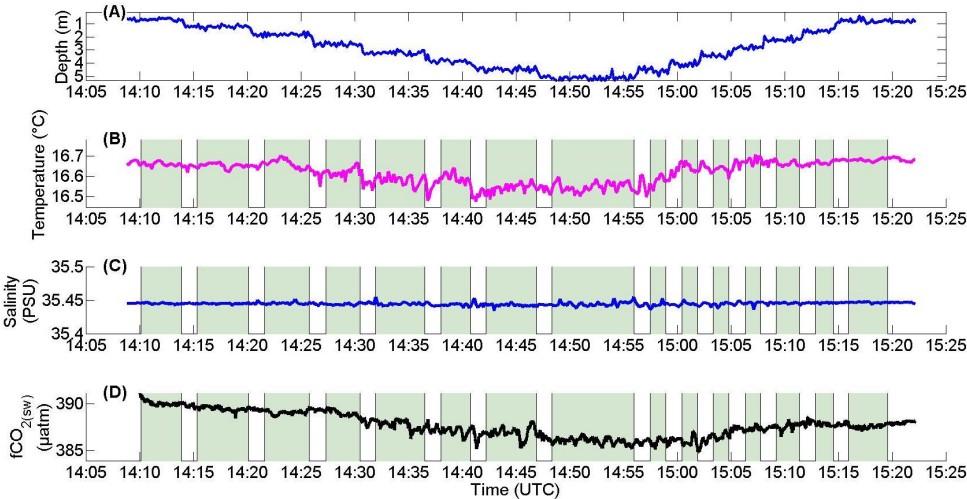

**Figure 6:** Time series measurements made during an NSOP deployment in the Celtic Sea on
30[th] July 2015. Data are 1 Hz depth (a), seawater temperature (b), salinity (c) and $fCO_{2(sw)}$ (d).
Data used for depth bin analysis (Section 2.6) is identified by a shaded background.
Depth-binned salinity and temperature data did not show any significant variability (Fig. 7a).
A slight temperature gradient was observed, with 0.15°C difference between 5 m and the
surface and a fairly constant reduction with depth (0.03°C per metre). The temperature profile
was similar for down and up casts, although some continued warming of surface waters was
evident in the up cast. The temperature measured by NSOP at 5.15 m depth agrees well with
the coincident temperature measured by the bow thermistor at 5.5 m (< 0.02°C difference)
(Fig. 7c). There is no evidence that the ship's thrusters/propellers disrupted the near surface
gradients.
We compare the NSOP temperature profile with thermistor readings from a series of sensors
on a mooring ~2.8 km away (0.3, 0.6, 1.5, 3.5 and 7 m depth). The vertical profile implied by
the NSOP deployment agrees with the mooring data (Fig. 7c), and corroborates the warming
of the upper few metres of the ocean observed during the deployment. The agreement
between these independent datasets suggests that it is unlikely that NSOP caused any
significant localized warming of surface waters. The mean difference between NSOP
temperature from discrete depths and the mooring sensors is 0.02°C. The surface data from
the NSOP up cast show less agreement with the mooring, with NSOP temperatures ~0.05 °C
lower than the 0.3 m and 0.6 m mooring sensors. During the profile the ship drifted ~1 km




from the start position of the profile and a further 0.2 km from the mooring. The small offset
between the NSOP surface temperatures and the mooring may be driven by horizontal
variability between the deployment and mooring locations. It is also possible that turbulence
mixed warm surface waters down into cooler sub-surface layers. Turbulence could have been
generated around the NSOP sampling frame or by an increase in wave-driven mixing when
the significant wave height increased at ~15:00 hrs UTC (Fig. 5a).

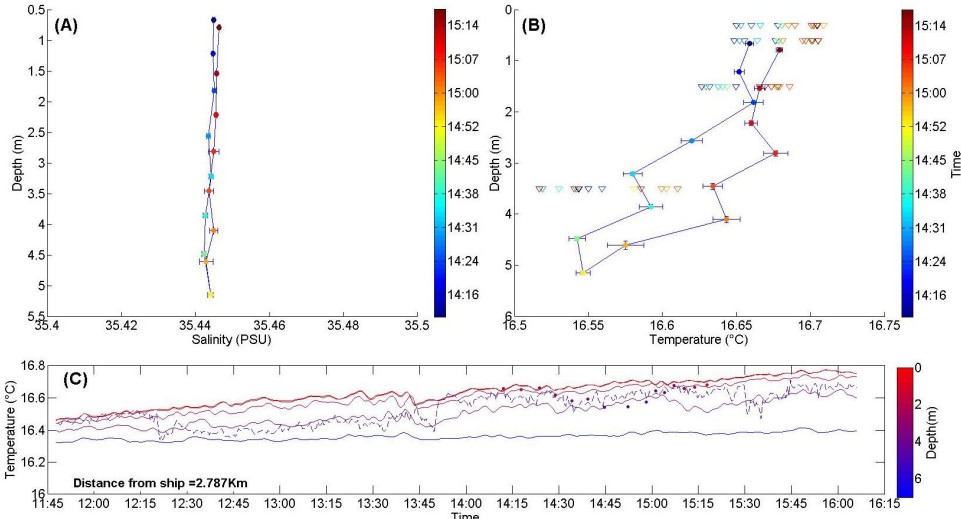

**Figure 7:** Salinity and temperature in the Central Celtic Sea on 30[th] July 2015. NSOP profiles
of salinity (a) and temperature (b) were derived using depth bins as described in Section 2.6.
Data points are coloured by sampling time. Vertical and horizontal error bars show two
standard errors of the mean in each depth bin. Coloured triangles in (B) are time-averaged
temperature for four depths (0.3, 0.6, 1.5 and 3.5 m) at the nearby Central Celtic Sea
temperature mooring (49.403°N, -8.606°E). (c) Timeseries of temperature at the mooring.
Dashed line is the underway temperature at 5.5 m from *RRS Discovery* (located 2.8 km from
the mooring). Coloured circles are binned temperature data from NSOP. Sample depth is
indicated by blue-red colour, where red is the air/sea interface.
Seawater density (Fig. 8a) was calculated using the salinity and temperature profile data (Fig.
7a &7b) and the 1983 Unesco equation of state (Millero and Poisson, 1981). As expected with
little variation in the salinity, changes in the density profile are dominated by temperature.
The down and up casts for $CO_2$ show excellent agreement below 2.5 m. Surface water (< 2 m)
$CO_2$ is 2-4 µatm higher than at 5 m (Fig. 8b). Elevated surface $CO_2$ could be explained by a
sustained flux from the atmosphere into a near surface stratified layer with inhibited deep




water exchange. Under this assumption a vertical gradient in seawater $CO_2$ would need to be
established shortly after the temperature gradient. Surface $CO_2$ is significantly different
between the down and up casts. The deepening of the surface stratified layer could explain the
more homogeneous $CO_2$ during the upcast. It is worth noting that in addition to physical
processes, plankton trapped within the surface layer could also modify the surface $CO_2$.

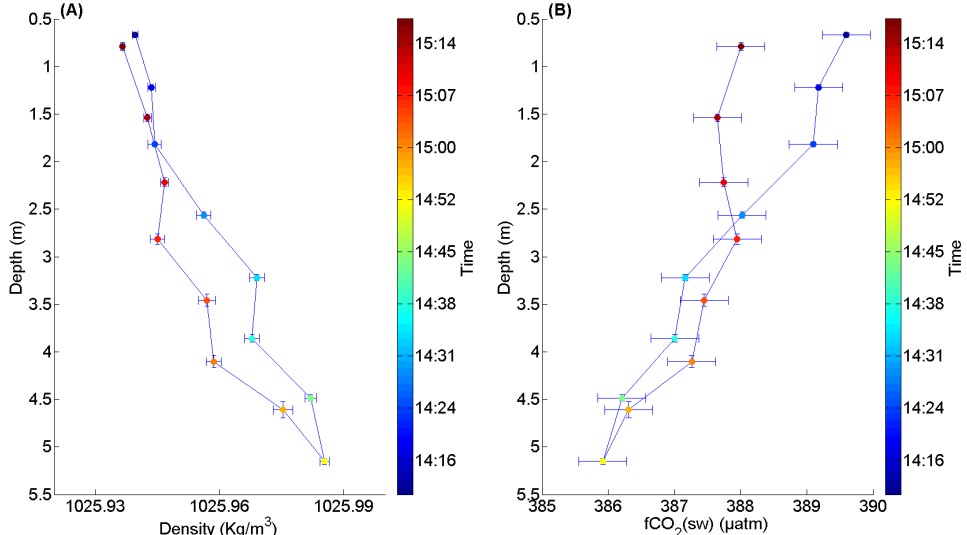

**Figure 8:** NSOP density (a) and $fCO_{2(sw)}$ (b) profiles from the Celtic Sea on 30[th] July 2015.
Data points are coloured by sample time. Vertical error bars correspond to two standard
errors of the mean in each depth bin. The horizontal error bars in (a) are two standard errors of
the mean, whereas in (b) they are the propogated error from the the binned measurements used
to calculate $fCO_{2(sw)}$.
To assess measurement accuracy the NSOP Liqui-Cel $CO_2$ system was compared against an
independent $CO_2$ system that had a showerhead equilibrator coupled to the ship's seawater
supply pumped from 5.5 m below the sea surface (Hardman-Mountford et al., 2008; Kitidis et
al., 2012). Technical issues meant that the underway $CO_2$ system installed on the *RRS*
*Discovery* was not functioning during the deployment detailed above. However during
deployments on the 19[th] and 20[th] July, the $fCO_{2(sw)}$ measured by NSOP close to the underway
intake depth agrees to within 3 μatm.



## 3.2 Coastal deployment

DMS profiles were collected on a small research vessel on 15[th] April 2014. NSOP was deployed within the Plymouth Sound at 12:00 hrs UTC and recovered 95 minutes later (Fig. 10). In the sheltered environment behind the breakwater the standard deviation in depth was ±0.10 m, smaller than observed during open ocean profiles. Seawater temperature and salinity demonstrate clear structure, with lower temperatures and higher salinities associated with sub-surface water. Two river estuaries (Plym and Tamar) converge and flow out to the open ocean through the Plymouth Sound. We likely observed a freshwater surface lens that was protected from wave-driven mixing and had been warmed over the course of the day. We used a different miniCTD during this deployment and were thus also able to collect fluorescence data (Fig. 10d).

Temperature profiles (Fig. 11a) show a sharp discontinuity in the downcast at ~5 m whereas in the upcast the thermocline had shoaled to ~3.5 m. The salinity profiles suggest similar mixing depths to the temperature profiles, with lower salinity water at the surface (Fig. 10b). Fluorescence increases with depth (Fig. 10c), but this is likely due to quenching of the phytoplankton photosynthetic apparatus at the surface. DMS concentrations reduce steadily with depth (Fig. 10d), which is likely explained by changes in DMS production and consumption rates by the biological community (Galí et al., 2013). The DMS profiles from the upcast and the downcast are very similar, with the largest difference at the very surface. A large difference in the surface-most data point can also be seen in the temperature data, and may reflect mixing with sub-surface waters due to the motion of NSOP or short time-scale variations in the physical environment.



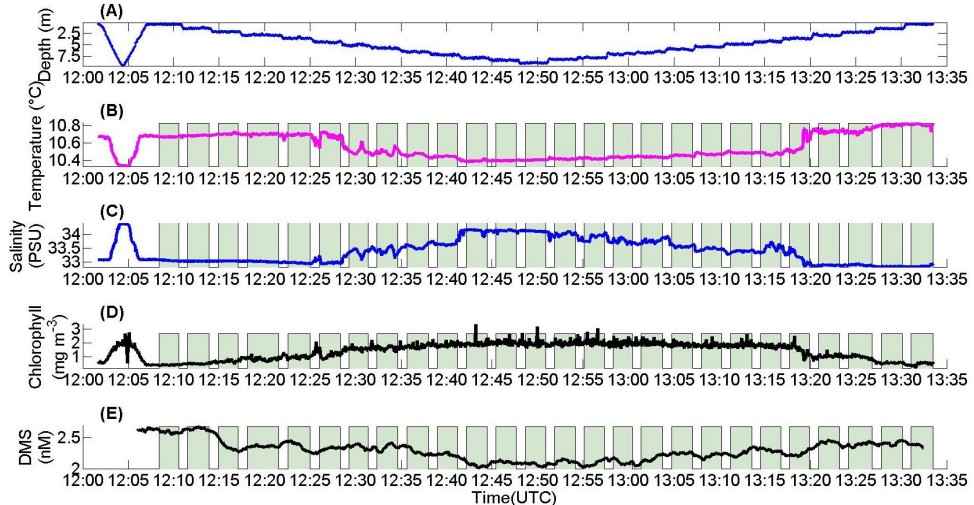

**Figure 9:** Time series measurements during an NSOP deployment in Plymouth Sound on 15[th]
April 2014: depth (a), temperature (b), salinity (c), chlorophyll fluorescence (d) and DMS$_{(sw)}$
(e). Data used for depth bin analysis (Section 2.6) is identified by a shaded background.

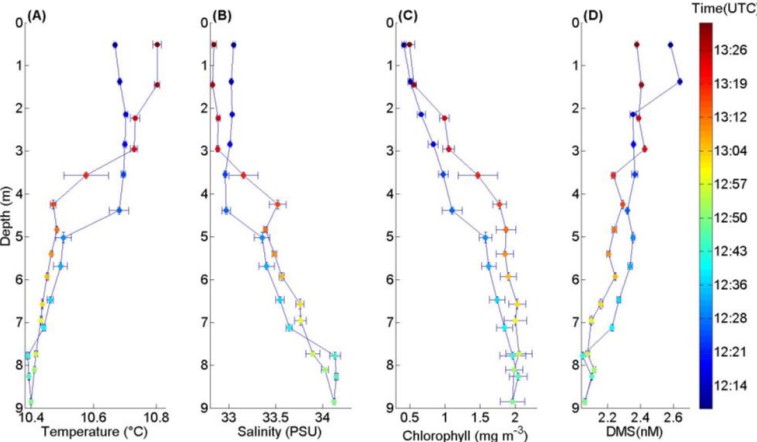

**Figure 10:** NSOP profiles collected in Plymouth Sound on 15[th] April 2014: temperature (a),
salinity (b), chlorophyll fluorescence (c), and DMS$_{(sw)}$ (d). Data are coloured by sample time.
Vertical and horizontal error bars are two standard errors of the mean (SEM) in each depth
bin.





## 4    Summary

This paper describes a Near Surface Ocean Profiler (NSOP) designed to measure vertical trace gas profiles near the air-sea interface. NSOP is unique in approach as its sampling frame is lowered from a buoy that rides the ocean swell, reducing relative motion of the frame and hence fluctuations in sampling depth. The NSOP design facilitates near surface ($< 0.5$ m) sampling, significantly improving the capability to resolve vertical gradients. Other benefits include the ability to sample away from ship-driven turbulence and the flexibility to make a large range of near surface measurements. The NSOP sampling frame houses the miniCTD and also has the capacity to incorporate additional sensors (e.g. dissolved oxygen and other measures of phytoplankton abundance and photosynthetic health). The ability to collect water from discrete depths facilitates the collection of near surface samples that require additional processing or take longer to analyse (e.g. TA, dissolved inorganic carbon, nutrients, the DMS-precursor DMSP, dissolved organic carbon). NSOP is highly versatile and can be used for continuous or discrete profiling. Further development could adjust winch pay out speed and enable continuous, high resolution depth profiles for slower response time measurements (e.g. $fCO_{2(sw)}$).

Near surface stratification in the upper few metres of the ocean due to temperature and salinity gradients is a well-documented phenomenon. The presence or absence of chemical and biological gradients within near surface stratified layers has been difficult to assess. NSOP is a platform with the capability to successfully resolve gradients in these near surface layers.

## Acknowledgements

We thank the captains and crews of  the *RV Plymouth Quest* and *RRS Discovery* for their assistance with deploying NSOP, Christopher Balfour and Dave Sivyer for maintenance of the Central Celtic Sea mooring near surface temperature sensors, Vassilis Kitidis for supplying underway $CO_2$ data and Burke Hales for advice concerning Liqui-Cel $CO_2$ measurements. This research was made possible by PML internal funding, a NERC funded studentship (NE/L000075/1), temperature sensors on the Central Celtic Sea mooring (NE/K002058/1) and by the NERC Shelf Sea Biogeochemistry pelagic research programme (NE/K002007/1). The *RRS Discovery* underway data was supplied by the Natural Environment Research Council.



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
