# Peer review of "A measurement system for vertical seawater profiles close to the"

_Ocean Science, 2017_

## Referee Comment (RC3) · Anonymous Referee #3 · 23 Apr 2017

This paper reports on a near surface profiler for sampling biogeochemical properties of seawater near the surface, where important biological and air-sea exchange processes take place, and where vertical characterization has proved difficult. The motivation in developing this profiling device would be to study the effect of stratification on the biogeochemistry at the air-sea interface.

There is no doubt that the authors have invested tremendous effort in developing and deploying the NSOP. However, I am a little disappointed with the scientific conclusions. In fact the article appears to be more of a technical description of an instrument, and therefore I think it might have been more appropriate for a more technical journal such as Methods in Oceanography. Therefore if the authors wish to publish this in OS, I suggest that they provide some further material on the scientific consequences of the NSOP. I do not think that this will require much effort, and although my major

revisions rating appear to be a little severe, I do not think it will require much effort. The current effort in their conclusions - "The presence or absence of chemical and biological gradients within near surface stratified layers has been difficult to assess. NSOP is a platform with the capability to successfully resolve gradients in these near surface layers." - is weak, and I have no doubt that given the list of authors here, a little more effort would provide much improved conclusions. For example: how well does the temperature-pCO2 relationship of 4.23% per degC hold? What are the global consequences for stratification on air-sea gas exchange?

---

## Author Comment (AC1) · 22 Jun 2017

We thank all 3 reviewers for their positive and constructive comments. We address their specific points below:

**Anonymous - reviewer 3**

**General comments**

This paper reports on a near surface profiler for sampling biogeochemical properties of seawater near the surface, where important biological and air-sea exchange processes take place, and where vertical characterization has proved difficult. The motivation in developing this profiling device would be to study the effect of stratification on the biogeochemistry at the air-sea interface. There is no doubt that the authors have invested tremendous effort in developing and deploying the NSOP. However, I am a little disappointed with the scientific conclusions. In fact the article appears to be more of a technical description of an instrument, and therefore I think it might have been more appropriate for a more technical journal such as Methods in Oceanography. Therefore if the authors wish to publish this in OS, I suggest that they provide some further material on the scientific consequences of the NSOP. I do not think that this will require much effort, and although my major revisions rating appear to be a little severe, I do not think it will require much effort. The current effort in their conclusions - "The presence or absence of chemical and biological gradients within near surface stratified layers has been difficult to assess. NSOP is a platform with the capability to successfully resolve gradients in these near surface layers." - is weak, and I have no doubt that given the list of authors here, a little more effort would provide much improved conclusions. For example: how well does the temperature-pCO2 relationship of 4.23% per degC hold? What are the global consequences for stratification on air-sea gas exchange?

We respectfully disagree with the reviewer. There is plenty of evidence of method-driven papers in Ocean Sciences (e.g. Saltzman et.al 2009, Hemming et.al 2017, Schneider-Zapp et.al 2014 and Arévalo-Martínez et.al 2013). A more detailed description of scientific results based on 4 cruises and a seasonal study is planned as a later publication, which will include discussions on the temperature/CO2 relationship and global fluxes. However, we agree that some of the results presented here can be discussed in more detail. To demonstrate how NSOP profiles may influence air/sea fluxes we have calculated how the fluxes change using the near-surface concentrations. We have added the following paragraph:

**We have adjusted the second paragraph of our conclusions (P20, L17):**

'Near surface stratification in the upper few metres of the ocean due to temperature and salinity gradients is a well-documented phenomenon. The presence or absence of chemical and biological gradients within near surface stratified layers has been difficult to assess. NSOP is a platform with the capability to successfully resolve gradients in these near surface layers. The data presented in this paper demonstrate that near surface gradients in trace gases can lead to substantially different fluxes depending upon the seawater depth that is used to calculate the flux. Assuming that the effect of temperature and salinity gradients on the flux can be accounted for using remote sensing methods (e.g. Shutler et.al 2016), then the change in flux is directly proportional to the change in  $\Delta C$ . In the case of the coastal DMS profile, a higher concentration (2.58±0.02 nM) was observed 0.5 m below the sea surface compared to concentrations at 5 m (2.36±0.03 nM). Assuming that the atmospheric concentration of DMS was negligible(a typical approach for DMS fluxes, see Lana et al.,

2011), computing the flux with the 5 m waterside concentration instead of the 0. 5m waterside concentration means the flux is underestimated by 9.3%. . In the case of the Celtic Sea CO2 profile, the concentration at 0.5 m (389.60  $\pm$ 0.36 µatm) was higher than at 5 m (385.92  $\pm$ 0.36 µatm). The atmospheric CO2 concentration was 398.1  $\pm$ 0.3 µatm, which means that the surface water was less undersaturated than implied by the seawater concentration at 5 m. Using the 5 m waterside CO2 concentration. The magnitudes of these concentration gradients are significant. However, such gradients (in magnitude and direction) do not persist for all hours of the day, under different environmental conditions and in all regions of the global ocean. A subsequent publication will discuss NSOP data collected during four cruises as well as the wider prevalence and implications of near surface CO2 gradients.'

**Maria Ribas Ribas – Reviewer 1**

**General comments**

The present paper under review for Ocean Science describes state-of-the-art technology to measure high resolution profile in the upper 5 m of the ocean. I appreciate the effort of research, develop and validation of the authors. Everyone working on R&D knows that behind these two examples profiles are a lot of trial-error and frustration. I also think the described technology fill a gap and it is really important and that it is adequate to the scope of the journal. I will recommend publication after some minor/moderate revision. I hope the comments help to improve the ms. I understand first author is PhD student and I congratulate him for the nice work .

Specific comments

Page 4-line 15 Define ID-

As this is the only occurrence of ID, we have changed ID to inner diameter

4-25 I am curious to know what the maximum wave height is and wind you deploy. Also applicable for my first comments on real live application.

This information is already in the text on Pg 5 L 21-23

5-4 What is the maximum distance?

In this case the maximum extended length of the crane arm was ~7m. Pg 5 L 3-4 changed to

"through a block on a fully extended crane arm of 7m to maintain this distance between NSOP and the ship."

5-12 unfortunately I can't access to supplement material. I would love to see the videos

5-22 What is the limitation of the deployment length? Battery? Can other deployments, ship operations happen at the same time, like CTD, so you have a concurrence profiles?

The information on the maximum deployment length is already in the manuscript on Pg 6 L 12-15. The fact that no other instrumentation can be deployed simultaneously is a pertinent point. Pg 4 L 28 changed to "NSOP was always deployed while the ship was on station and not at the same time as other overboard deployments."

**6-10 Please check figure order of appearance, suddenly here we found Fig. 9.**

Reference has been removed and a reference to this part of the text has been included in the caption of figure 9.

**7-21 In text and figure, unify use of litre with capital L**

All instances have been changed .

**7-29 How was the pressure inside the equilibrator measured?**

All pressure measurements were made with the Licor analyser, in close proximity to the equilibrator. Unfortunately there is no way to measure the pressure internally without compromising the membrane. We are confident that the equilibrator pressure is 0.4 kpa above ambient. Changed p7 L 26 to 'The continuous gas flow through the system caused a small 0.4 kPa pressure increase in the Licor measurement cell, this was in good agreement with a similar observation by Burke Hales (0.5kpa > ambient pressure; Personal communication).'

**9-1:4 Could you provide more details of the membrane ( $\mu$ m...) –**

The membrane material, total surface area and internal liquid and gas side volumes are now included. Pg 9 L 1 change to "We used a polypropylene membrane equilibrator (Liqui-Cel, model 2.5x8) with liquid and gas volumes of 0.4 L and 0.15 L and a surface area of 1.4 m2. Due to its large surface area to volume ratio and membrane porosity (50%), the Liqui-Cel expedites gas transfer and efficiently achieves equilibration"

*Figure 3 caption: Could you add legend (nice to understand the figure without the need to read first). What is LPM in the x-axis?-*

Legends are discouraged by the ocean science journal. LPM has been changed to L min-1 and ml changed to mL throughout.

10-16 what is m/z –

Mass is now defined as m and charge as z.

10-19: two points in the reference 11-12: wrong use of () in the reference -

This has been corrected.

Figure 4: what is the magnitude of the change? For example from 400 to 1000 ppm or 400 to 450 ppm (that will make a different, right?)? –

The change was from 350 to 400 ppm for  $CO_2$  and 0 to 2 nmol L-1 for DMS. The absolute magnitude of the change does not matter for the calculation of response time so long as the change is large enough to be clearly detected by the instrument.

*I miss a table of comparison of discrete and continuous operation. Also another one of the sensor use with the accuracy/frequencies... to have a quick overlook of the system. –*

The two modes of deployment are not directly comparable and there is insufficient information to justify a table.

12-23 Probably not need to say the SOP# -

We feel it is helpful to retain the reference to the SOP as the document is ~200 pages.

In all figures, A), B) C) in the figures are capital but in the captions are not. Please unify.

The capitalisation in the figures has been changed.

**14-13 How much of the unsaturated?**

The atmospheric CO2 concentration is now included. P14 L 13 changed to 'Seawater temperature was 16.61± 0.06 °C. At 14:20 hrs (UTC) fCO2(atm) was 398  $\mu$ atm andfCO2(sw) was 389  $\mu$ atm at 0.67 m meaning the ocean was undersaturated with respect to the atmosphere. The temperature and seawater CO2 were the expected magnitude for summer in the Celtic Sea (Frankignoulle and Borges, 2001).'

15-15 Could you provide a bit more detail about this mooring? Maybe a map with location site and the mooring will be helpful for readers not familiar with the area.

A map of both deployment sites in now in the supplement. Additional information about the mooring can be found in the cruise report

https://www.bodc.ac.uk/resources/inventories/cruise\_inventory/reports/dy033.pdf.P15 L15 changed to 'We compare the NSOP temperature profile with thermistor readings from a series of Sea-Bird Scientific (SBE 56) sensors (0.3, 0.6, 1.5, 3.5 and 7 m depth) mounted on a nearby temperature chain moored ~2.8 km away (49.403°N, -8.606°E) from the deployment site'.

*Figure 6 and 9 are quite confusing, as depth is plot with time instead as usual oceanography profile way. In a related note, what is the different info from figure 9 and 10?*

These plots demonstrate that the data is collected continuously at high frequency rather than discrete samples. We have shown the time series data here as high frequency seawater  $pCO_2$  data is rarely presented, so showing how this varies is important. In addition, this paper focuses on a new method and we want to be absolutely clear of the data processing steps that are required rather than 'rush to the final plot'.

16-1:6 Can the drifting from ship cause turbulence/mixing? Would be possible to measure turbulence within the NSOP?

NSOP may create a small amount of local turbulence but we did not see any evidence of this in our profiles. It would be possible to measure turbulence from NSOP and we have added turbulence sensors to the list of possible additional sensors on P 19 L 9.

**17-2 When you talk about significantly different I expect a statistical test.**

As the errors on the  $CO_2$  measured by NSOP and the underway system are two standard errors, the fact they don't overlap indicates they are statistically different at the 95% confidence interval. To be absolutely clear, P17 L2 has been changed to 'A paired t-test showed that the fCO2 measured in the surface bins on the downcast and upcast are were significantly different (p = <0.001).'

**17-1:5 This paragraph is really important and key message of paper so I will like to have more discussion on it. What is the role of sea surface microlayer?**

The sea surface microlayer was not discussed in detail as it could not be sampled with NSOP, we agree that it should be mentioned. P 17 L1 Added 'Trace gas concentrations may also be different in the sea surface microlayer but sampling that close to the surface is beyond the capabilities of NSOP. Complimentary measurements of the sea surface microlayer could be made using other state of the art purpose built sampling platforms such as the Sea Surface Scanner (Ribas –Ribas et.al., 2017).'

What is the implication of the calculations of flux as normally do it from 5 m? ...

Refer to our response to anonymous reviewer 3.

17-18:19 This comparison with underway CO2 is also really important. Can you provide more detail? I will think that ship disturbance will have more influence of underway system... What is RMS? Why we should care about NSOP if they give similar results of usual pCO2 instrument? I really think it is important, do not take me wrong, I just think a bit more discussion will be good

The original manuscript was unclear so we have improved the wording:

We have changed P17 L 18 to:- 'However during a deployment on the 19th July 2015, the  $fCO_{2(sw)}$  measured by NSOP at 5 m agreed well with independent measurements from the underway system, difference = 1.7+/- 4.18 µatm, ). The agreement between the two systems is in line with previous intercomparisons (Kortzinger 2000; Ribas-Ribas 2014).

**Anonymous – reviewer 2**

**General comments**

The authors presented a new sampling technique to sample and measure the vertical profiles of physical and chemical parameters in the subsurface layer of the ocean (<10 m). This indeed improves the very surface layer sampling for chemical tracers. The paper is overall well written with the exception of a few confusing sentences. Some of the figures present redundant information, and could be removed. However, the main problem of the paper, I feel, is lack of discussion on how the whole technique may impact the estimation of air-sea exchange. Does it really matter that we need to sample the very subsurface layer (e.g.,  $\sim$ 1m) of the seawater to obtain a very accurate flux

estimate? Or it may be good enough to sample 4m below the sea surface as the traditional sampling system?

In response to a similar comment from Reviewer 3, we have added a paragraph on this subject.

**Specific comments**

Pg2, lines 22 – 30: Authors discussed near surface stratification due to certain physical processes here. It is worth to note and mention that, people have been using CFC-11 to correct physical effect on estimated air-sea fluxes (see Lobert et al., 1995; Butler et al., 2016; Yvon-lewis et al., 2004; Hu et al., 2013).

This section discusses the calculation of air/sea flux using estimates of gas transfer velocity combined with global databases of trace gas concentrations. This approach to calculating air/sea flux is commonly-used in Earth System models. Near surface gradients may impact upon the calculation of air/sea fluxes.

We are not claiming that other approaches (such as the use of CFCs, which incorporate the implicit effects of physical forcing on gas exchange) would be influenced by near surface gradients. However, discussing the CFC method would be a deviation from the narrative of the text so we have not added anything to this effect.

Pg12, lines 7 – 9: "The delay between a bucket switch and CO2 change in the Licor was timed at 138s...". This is confusing. Does this delay include CO2 response time in the equilibrator and the time from sampling to the equilibrator? If it is, define it. Also, this sentence seems out of place here.

P12 L7 We have changed the wording and structure of this section to improve clarity:

'We used different approaches to assess the delay between instantaneous miniCTD measurements and water arriving to the ship for analysis. The delay between seawater entering the inlet and reaching the equilibrator was calculated as 114 s using the internal volume of NSOP tubing (0.5 in ID, 54 m length) and a seawater flow rate of 4.15 L min-1.Delay correlation analysis between the NSOP miniCTD temperature sensor and a second sensor positioned at the entrance to the equilibrator gives a similar delay of 112 s. Note that the total delay of the system is greater because it also includes the time that equilibrated gas takes to reach the Licor. We determined the total delay by moving the seawater inlet quickly between two buckets with distinctly different CO2 concentrations and timing how long it took for the signal to be detected by the Licor (139 s; Fig. 4).'

Pg12, lines 21 – 25: list equations to be clearer.

The equations are commonly used by the community and are well detailed in (Dickson 2007). We feel this is sufficient.

Pg 12, lines 26 – 28: redundant information.

**Removed**

*Fig. 5: I don't think this figure provides extra information or value other than the sentence described in Pg 11, line21. So, you may consider remove it.*

The reference to this figure remains in the text but the figure has now been moved to the supplement.

*Figs. 6 & 7: Redundant. Recommend to remove fig. 6. Figs. 9 and 10: redundant information. Consider remove fig. 9*

Please see earlier response to Ribas-Ribas.

Also, fig. 7 looks a little messy with depth contours.

The lines in Fig. 7c are not depth contours. They are a time series of temperature for sensors at different depths. We have added additional detail to the figure legend to clarify this.

*Pg. 16, line 18: Density was not used in the later discussion. I am not sure why you want to mention and discuss density profiles here. Consider remove it from the text and figure.*

Density determines the stratification and is affected by both salinity and temperature. We think it is necessary to include density as it is more useful to the reader than just plotting temperature.

Pg. 17, lines 3 - 5: why do fco2 profiles show the largest difference in the surface, and not in the layer where the temperature showed the largest difference (4.5 - 2 m below water)? The explanation given by the authors is not convincing. Since the amount of co2 outgassing due to surface seawater warming (during upcast) can be calculated using temperature, salinity and solution of CO2, it is not hard to estimate how much the difference between down cast and up cast was due to physical effect, and how much was due to biological influences.

We are a bit confused by this comment. Figure 8 shows that there is a gradient in  $fCO_2$  down to a depth of 5 m. The temperature effect on  $fCO_2$  is already accounted for - the profile we show is seawater  $fCO_2$  after correction for *in situ* temperature.

Pg. 17, lines17–19: 3 uatm seems ahuge difference considering the variation off co2 observed in the subsurface layer. Did the authors consider the different response time in two different equilibrators? Why are their measured values so different?

We apologise - our original text is misleading. We said that " $CO_2$  agrees to within 3 uatm" but the mean difference is actually 1.7+/- 4.18 uatm. This is very similar to previous comparisons between shower and membrane equilibrators (Hales et.al,. 2004). This is also close to the (Kortzinger et.al,. 2000) comparison of 1 uatm, which used very similar equilibrator setups. As detailed in our response to reviewer Ribas-Ribas, we have changed the paragraph on P17, L 17-19 to make this clearer. Also, note that we do consider the response time in the equilibrators (see text on P10, L1-14).

*Pg. 18, lines 15 – 16: confusing. To me, the increased fluorescence is likely due to phytoplankton located at the bottom or below the subsurface layer. –*

We disagree. It is well established that phytoplankton fluorescence becomes quenched in the very near surface layers and does not suggest a change in phytoplankton concentration with depth. (Serra et.al 2007, Gibb et.al 2000 and Smyth et.al 2004).